# Comparison of Different Nutritional Assessment Tools in Detecting Malnutrition and Sarcopenia among Cirrhotic Patients

**DOI:** 10.3390/diagnostics12040893

**Published:** 2022-04-03

**Authors:** Mirabela-Madalina Topan, Ioan Sporea, Mirela Dănilă, Alina Popescu, Ana-Maria Ghiuchici, Raluca Lupușoru, Roxana Șirli

**Affiliations:** 1Department of Gastroenterology and Hepatology, “Victor Babeș” University of Medicine and Pharmacy, 30041 Timisoara, Romania; mirabelamadalina1990@gmail.com (M.-M.T.); isporea@umft.ro (I.S.); alinamircea.popescu@gmail.com (A.P.); anamaria.ghiuchici@gmail.com (A.-M.G.); raluca_lupusoru@yahoo.ro (R.L.); roxanasirli@gmail.com (R.Ș.); 2Advanced Regional Research Center in Gastroenterology and Hepatology, “Victor Babeș” University of Medicine and Pharmacy, 30041 Timisoara, Romania; 3Center for Modeling Biological Systems and Data Analysis, Department of Functional Science, “Victor Babeș” University of Medicine and Pharmacy, 30041 Timisoara, Romania

**Keywords:** malnutrition, sarcopenia, liver cirrhosis, nutritional screening tools, anthropometric measurements, handgrip strength, skeletal muscle index

## Abstract

Malnutrition and sarcopenia are common complications of liver cirrhosis. This study compares the performance of different nutritional assessment techniques in detecting malnourished patients. Data from 156 patients with liver cirrhosis were collected. We assessed the nutritional status of these patients according to: Subjective Global Assessment (SGA); Royal Free Hospital-Nutritional Prioritizing Tool (RFH-NPT), skinfold thickness (TSF), mid-upper arm circumference (MUAC), mid-upper arm muscle circumference (MUMC), handgrip strength (HGS), body mass index (BMI), and skeletal muscle index (SMI) evaluated by Contrast-Enhanced Computed Tomography (CT). According to EWGSOP2 criteria, combining low HGS with low SMI, the prevalence of malnutrition/sarcopenia was 60.2%. RFH-NPT, MUAC, MAMC, and HGS were excellent tests for detecting malnourished patients. Combining RFH-NPT with MUAC or MUMC increased diagnosis accuracy, AUC = 0.89, *p* < 0.0001. Age, Child-Pugh class C, albumin level, vitamin D deficiency, male gender, and alcoholic etiology were significantly associated with malnutrition. In conclusion, the prevalence of malnutrition among patients with cirrhosis was relatively high. Our study highlights the potential use of a simpler and inexpensive alternative that can be used as a valuable tool in daily practice, the combination between RFH-NPT and MUAC.

## 1. Introduction

Malnutrition is a frequent but often overlooked complication in patients with advanced liver disease, and it is a significant prognostic factor for morbidity and mortality [1]. The reported prevalence of malnutrition (undernutrition) in cirrhosis ranges from 23–60% [2]. It is defined as a change in mental and physical functions due to altered body composition and cell mass, leading to poor clinical outcomes and impaired quality of life [3]. Sarcopenia is a main component of malnutrition [4], and it is defined as a generalized loss of skeletal mass, strength, and function [3].

Although these terms have different definitions, the new American Association for the Study of Liver Diseases (AASLD) practice guidance on managing malnutrition, frailty, and sarcopenia in patients with liver cirrhosis acknowledges that sarcopenia and malnutrition are interrelated. Thus, we can describe a patient with cirrhosis presenting severe muscle wasting as malnourished or sarcopenic [5].

Most recent guidelines have highlighted the importance of identifying the presence of malnutrition in patients with liver cirrhosis [5,6]. However, it is a challenge in clinical practice because even if there are various traditional tools to determine nutritional status, and most have not been validated in cirrhotic patients.

Nutrition assessment tools are objective measures intended to diagnose malnutrition. Global assessment tools such as Subjective Global Assessment (SGA) and Royal Free Hospital-Nutritional Prioritizing Tool (RFH-NPT) are standardized batteries of questions and physical examination techniques, performed at the bedside. SGA is commonly used to assess nutritional status in patients with liver cirrhosis [7]. Even if there are studies that demonstrate the correlation between malnutrition assessed by SGA and mortality in cirrhotic patients [8,9], the use of SGA has some limitations, such as underestimating the prevalence of sarcopenia [10]. RFH-NPT is a tool developed specifically for cirrhosis, and it is an independent predictor of clinical deterioration and transplant-free survival [11].

The European Working Group on Sarcopenia in Older People (EWGSOP2) Guideline on sarcopenia 2019 [12] states that sarcopenia should be suspected in cases with muscle strength loss. One of the proposed tools for the first-line assessment of muscle strength is handgrip strength (HGS), which has proven to be reliable in evaluating sarcopenia in cirrhotic patients, predicting patients’ poor outcomes and mortality [8,13]. The second step suggested for diagnosing sarcopenia is measuring muscle quantity or mass. The most specific method to quantify muscle loss is the skeletal muscle index (SMI), evaluated by Contrast-Enhanced Computed Tomography (CT) [6]. CT is frequently used in cirrhotic patients for routine screening for hepatocellular carcinoma (HCC), so it can also be used to assess sarcopenia, but it has some limitations regarding radiation exposure and costs.

Body mass assessment can also be performed by anthropometric measures, such as mid-upper arm circumference (MUAC), mid-upper arm muscle circumference (MUMC), and triceps skinfold thickness (TSF), which are easily and rapidly performed. These procedures are inexpensive but are affected by fluid retention. Both MUMC and TSF have prognostic mortality values among cirrhotic patients [9].

The study aims to compare the performance of different nutritional assessment techniques in detecting malnourished cirrhotic patients. The reference method is a combination between low SMI and low handgrip strength according to EWGSOP2 criteria [12]. To use it in clinical practice, we wanted to find the most sensitive, cost-efficient, and easily repeatable method to identify malnourished/sarcopenic patients. As a secondary aim, we evaluated the incidence of malnutrition among cirrhotic patients from our department.

## 2. Materials and Methods

### 2.1. Population Selection and Study Design

We have conducted a prospective, observational study performed in a tertiary Department of Gastroenterology and Hepatology of the Emergency County Hospital, Timisoara, Romania, from January 2019 to December 2020 on 156 patients with liver cirrhosis.

The diagnosis of liver cirrhosis was based on physical examination, laboratory tests, abdominal ultrasound, ultrasound-based elastography, upper endoscopy, and radiological evidence. The severity of cirrhosis was assessed by the Child Pugh’s score [14] and by the Model for End-Stage Liver Disease (MELD) score [15]. All the tests were performed during the same admission, except for Contrast-enhanced CT scans, which were performed at most one month from baseline.

Among the 631 patients with liver cirrhosis admitted in our department during the study time frame, only 156 fulfilled all the inclusion criteria and were included in the final analysis. The inclusion criteria were: patients with liver cirrhosis, age greater than 18 years, and a diagnostic reference standard method (Contrast-enhanced CT). The exclusion criteria were patients with hepatorenal syndrome, with coexisting Human Immunodeficiency Virus, tuberculosis, septicemia, chronic renal failure, inflammatory bowel disease, hepatocellular carcinoma, or other malignancies.

The study protocol was designed following the Helsinki Declaration of 1975. Informed consent to participate in the study was obtained from every patient. The local Ethical Committee from “Victor Babes” University of Medicine and Pharmacy Timisoara approved the study number 41/10.12.2018

Nutritional assessment was done using malnutrition screening tools (SGA, RFH-NPT) and anthropometric measurements (Triceps skinfold, MUAC, handgrip strength, BMI, skeletal muscle index). We applied the EWGSOP2 criteria [12] and defined sarcopenia/malnutrition as low muscle strength determined by handgrip strength and low skeletal muscle index evaluated by contrast-enhanced CT. We used their combination as our reference method.

### 2.2. Nutritional Assessment Tools

The *SGA* was the first modality used for nutritional assessment. It is a simple tool that uses information obtained by clinical history and physical examination and classifies the nutritional status of patients into three categories: well-nourished, moderately, or severely malnourished [16].

The European Society for Clinical Nutrition and Metabolism (ESPEN) guidelines recommend the *RFH-NPT* as an excellent method to identify the risk of malnutrition in patients with liver disease. It is based on the information given by the patient regarding oral intake, recent weight loss, fluid overload, and BMI. Patients are then categorized as being at low (score 0), moderate (score 1), or high risk (score 2–7) for malnutrition.

### 2.3. Anthropometric Measurements

*Body mass index* (*BMI*) was calculated in all patients using the equation weight (kg)/height (m)^2^. Since most patients with liver cirrhosis have ascites and/or edema, dry weight (dry BMI) [6] was calculated by subtracting a percentage of the actual weight, based upon the severity of ascites: 5% if mild, 10% if moderate, and 15% if severe, adding another 5% if the patient had bilateral pedal edema. The patients were classified as underweight if BMI < 18.5 kg/m^2^, normal if BMI was 18.5–24.9 kg/m^2^ and overweight if BMI > 25 kg/m^2^.

*MUAC*: the measurement was performed on the right arm, midway between the scapular spine and the tip of the olecranon process, with a measuring tape placed perpendicular to the length of the arm at the marked location. MUAC was measured in centimeters. The average of three measurements was taken into consideration for further analysis.

*TSF*: the measurement was performed on the mid-line of the posterior surface on the arm, halfway between the acromion and olecranon process, with the arm held freely to the side of the body. It was measured in millimeters using a Lange metal caliper. The average of three measurements was taken into consideration for further analysis. Once MUAC and TSF measurements were completed, *Mid Upper Arm Muscular Circumference* was calculated using the following equation [17]:MUMC (cm)=MUAC (cm)−TSF (mm)∗0.314

MUAC, MUMC, and TSF were classified according to the reference values on Frisancho percentile tables [18], and after that, according to Blackburn and Harvey [19], they were classified into two groups: between 5th and10th percentile—mild malnutrition and <5th percentile—severe malnutrition.

*HGS*: dominant handgrip strength was measured using a Jamar dynamometer. The patient was examined while sitting down, having the arm along the body and the elbow flexed at 90°. For the dorsal position, the elbow was supported, and the head was at 30°. Each patient performed the test three times using the dominant hand, pausing 10–30 s between the tests. All values were recorded in kilograms. Cut-offs values used were: HGS < 27 kg for men and HGS < 16 kg for women [12].

*SMI*: CT images for cross-sectional skeletal muscle mass assessment were analyzed at the level of lumbar three vertebra (L3) by a single observer, using National Institutes of Health ImageJ (NIH ImageJ) software. Standard attenuation values considered for muscle tissue varied from 29 to 150 Hounsfield units (HU). The cross-sectional areas obtained were normalized for patient height. Skeletal muscle index was obtained, which is expressed as the cross-sectional muscle area/height^2^. An experienced radiologist performed the measurements. Cut-off values used for the presence of sarcopenia were: SMI < 50 cm^2^/m^2^ for men and 39 cm^2^/m^2^ for women [6].

### 2.4. Statistical Analysis

The statistical analysis was realized using MedCalc software for Windows (MedCalc Software, version 19.3.1, Ostend, Belgium). Categorical data were described as numbers and percentages. Continuous data were characterized as mean and standard deviation and when needed, data were categorized (risk analysis). Skewed data were referred to as median and interquartile-line. Pearson correlation, with its special case point-biserial correlation and Phi correlation for dichotomous data, Spearman’s rho and Kendall’s tau-b were used for correlation analysis of categorical data. The Point-Biserial Correlation is a special case of the Pearson Correlation and is used when measuring the relationship between a continuous variable and a dichotomous variable, or one that has two values. Phi represents the correlation between two dichotomous variables. As with the point-biserial, computing the Pearson correlation for two dichotomous variables is the same as the phi. The phi is a nonparametric statistic used in cross-tabulated table data, where both variables are dichotomous. A receiver operating curve analysis (ROC) was generated for the performance of every test in diagnosing malnutrition, and area under the curve (AUC), sensitivity, specificity, positive predictive value (PPV), negative predictive value (NPV), test accuracy, as well as optimal thresholds were calculated according to Youden Index. The DeLong test was used to compare the ROC curves. When needed, two continuous variables were combined and categorized, in order to find a better performance, into a new variable named “sarcopenia”, according to their cut-off values for malnutrition.

Kappa coefficient (k) of agreement, concordance correlation coefficient, and Cronbach’s alpha between each method were assessed. In order to do this analysis, the continuous variables were categorized, for example, for BMI we made a new variable, with BMI > 18.5 kg/m^2^ as malnutrition (yes), and BMI >18.5 kg/m^2^ without malnutrition (no); the same was done with SGA, MUAC, MUMC, and TSF, according to their cut-off values of malnutrition [9,12,16,17]. A 5% significance level was considered. Logistic regression analysis was used for identifying the predictors of malnutrition.

## 3. Results

### 3.1. Patients’ Characteristics

In total, 156 patients fulfilled the inclusion criteria and were included in the analysis. The mean age of the studied patients was 61.8 ± 8.7. The male gender was predominant at 61.5%. Regarding etiology, 57.1% had alcoholic cirrhosis, 25.6% hepatitis C virus (HCV) cirrhosis, 11.5% hepatitis B virus (HBV) cirrhosis, and 5.8% other etiologies. According to the Child-Pugh Classification: 21.8% were A-class, 39.1% were B, and 39.1% were C. Table 1 shows the baseline characteristics of the study population.

### 3.2. Prevalence of Malnutrition/Sarcopenia

The prevalence of malnutrition/sarcopenia in the cohort according to the standard cut off values of each method was: SGA 101/156 (64.7%), SMI 108/156 (69.23%), HGS 98/156 (62.8%), RFH-NPT 77/156 (49.3%), MUAC 75/156 (48.0%), TSF 54/156 (34.6%), MUMC 54/156 (34.6%), dry BMI 13/156 (8.3%), BMI 4/156 (2.5%), *p* < 0.0001.

Even it was not the focus of our study, we have also calculated sarcopenic obesity, which was defined as those patients with concurrent sarcopenia and overweight or obesity (BMI > 25 kg/m^2^). The prevalence of it in our cohort of patients was 27/85 (31.7%).

According to the EWGSOP2 criteria, when combining low HGS with low SMI, the prevalence of malnutrition/sarcopenia in our overall cohort was 60.2% (94/156). Moreover, 86/122 patients (70.4%) in the decompensated group had malnutrition, while only 8/34 patients (23%) were malnourished in the compensated group.

### 3.3. Malnutrition-Associated Factors

In the univariate analysis, we found out that age over 60 years, Child-Pugh class C, low serum albumin level, vitamin D deficiency, male gender, and alcoholic etiology were associated with malnutrition (undernutrition), assessed by the SMI and HGS, as shown in Table 2. Patients with Child-Pugh class C had a 3.5 times higher risk for malnutrition than those with class A or B, while male patients had a 3.4 times higher risk of malnutrition than women.

### 3.4. Comparison between the Used Methods

All the nutritional assessment methods were correlated with the presence of malnutrition. The correlation coefficients were as follows: RFH-NPT, r = 0.63, *p* < 0.0001; HGS, r = 0.55, *p* < 0.0001; MUAC, r = 0.49, *p* < 0.0001; MUMC, r = 0.45, *p* < 0.0001; SGA, r = 0.44, *p* < 0.0001; TSF, r = 0.23, *p* = 0.003; dry BMI, r = 0.22, *p* = 0.005; BMI, r = 0.14, *p* = 0.06.

The diagnostic performance of the investigated nutritional assessment tools, considering SMI evaluated by CT and HGS as a reference method, are presented in Table 3. RFH-NPT, MUAC, and MUMC alone performed the best, with AUROCs of 0.86, 0.81, and 0.79, respectively, *p* = 0.03.

We furthered our analysis to discover if combining several nutritional assessment tools would improve accuracy. The diagnostic accuracies of combined assessment tools are presented in Table 4. All *p*-values were <0.0001.

Besides the diagnostic accuracies presented in Table 4, a model including RFH-NPT and MUAC or MUMC was generated, and it had the highest diagnosis accuracy, AUC = 0.89, *p* < 0.0001.

When evaluating the agreement between different nutritional assessment tools (Table 5), we found that the strongest agreement was observed between SMI + HGS and RFH-NPT, k = 0.62, *p* < 0.0001, followed by SMI + HGS and HGS alone, k = 0.55, *p* < 0.0001.

## 4. Discussion

The most objective and reproducible analysis of muscle mass is computed tomography image analysis at the level of the L3 vertebra [20]. Recently, the European Association for the Study of the Liver (EASL) [6] suggested the following cut-offs: SMI < 50 cm^2^/m^2^ in men and <39 cm^2^/m^2^ in female patients. In our cohort, by applying the EASL proposed cut-offs and the EWGSOP 2 criteria, sarcopenia was diagnosed in 60.2% of the patients, similar to the results presented by Bunchorntavakul C et al. in a recently published review article [2].

The main limitation of SMI measured by CT, besides the radiation exposure and costs, is the complexity of the measurement technique of the cross-sectional abdominal muscle area that requires radiological expertise and time and specialized software. This primarily impacts its use in daily clinical practice as a repeatable method. This is one of the reasons why we tried to identify a low-cost and easily reproducible nutrition assessment method, with similar performance to CT.

BMI and dry BMI were insensitive to identify malnourished patients. Only 2.5% were malnourished according to BMI and only 8.3% according to dry BMI vs. 60.2% according to the reference method. These values were smaller than those observed by Hassan et al. [21], who reported that 25.7% of the patients were malnourished according to BMI, but similar results were obtained in a more recent study by Nunes et al. [22], who reported that 8% were malnourished according to BMI.

Tandon et al. [23] also observed that SGA can only identify malnourished patients. In our study, the sensitivity of SGA was relatively high, 81.9%, but with low specificity, 61.2%, and an AUROC of 0.71. There was moderate concordance between sarcopenia determined by SMI + HGS and SGA, k = 0.44, *p* < 0.0001.

On the other hand, RFH-NPT, a screening tool specifically developed for patients with advanced liver disease, had the strongest agreement with the reference method, k = 0.62, *p* < 0.0001, and the best diagnostic performance, with an AUROC of 0.86, *p* < 0.0001. In a series of 148 patients, RFH-NPT was identified as an independent predictor of clinical deterioration and transplant-free survival [11]. Moreover, in a recent study, Georgiou et al. [24] compared the performance of eight screening tools in detecting malnutrition in cirrhotic patients. RFH-NPT and Liver Disease Undernutrition Screening Tool (LDUST) were the only screening tools that proved reliable in detecting malnutrition in patients with cirrhosis. Our study validates the performance of this nutritional assessment tool in cirrhotic patients.

According to MUAC and MUMC, 48% and 34.6% of patients were malnourished, respectively. The predictive value of these nutritional assessment tools as compared with SMI + HGS was outstanding, with AUROCs of 0.81 and 0.79, respectively, with a moderate correlation with SMI and HGS results (k = 0.47 for MUAC and k = 0.39 for MUMC, both *p* < 0.0001). Similar results were obtained by Tandon et al. [23].

Regarding TSF, it has been demonstrated that it has a prognostic value for mortality among cirrhotic patients but with lower prognostic power than MUMC [6]. In our study, TSF had a low diagnostic performance as compared with SMI + HGS, with an AUROC of 0.63 and weak agreement k = 0.20, *p* < 0.0001.

Skeletal muscle contractile function assessed by HGS only identified 62.8% of our patients as malnourished. The method had good results compared to the combination SMI + HGS findings (k = 0.55, *p* < 0.0001) and had a good predictive value (AUROC of 0.78, *p* < 0.0001). Analogous results were obtained in a recent study conducted by Tapper et al. [25], where HGS correlated with several findings, the strongest overall correlation being observed with skeletal muscle area (r = 0.641, *p* < 0.00).

In order to achieve better accuracy, we combined different nutritional assessment tools. A model made with RFH-NPT combined with MUAC or MUMC had the greatest diagnosis accuracy to identify sarcopenia, with an AUROC = 0.89, *p* < 0.0001. Considering that MUMC demands both MUAC and TSF measurements, the combination between RFH-NPT and MUAC is more rapid and easily repeatable for the follow-up of the patients.

We also observed that the factors associated with malnutrition were: age, Child-Pugh score, especially Child-Pugh class C, low albumin values, vitamin D deficiency, male gender, and alcoholic etiology. Published studies showed that patients with alcoholic liver disease are more likely to have poor nutrition as compared to other etiologies of cirrhosis [1] and so do patients with Child-Pugh B or C [26]. Given the fact that most of our patients were Child-Pugh B and C, and that the alcohol abuse was the most common etiology (57%), the high prevalence of malnutrition in our cohort can be explained. A higher prevalence of sarcopenia in male vs. female patients with cirrhosis, of 2:1, was also observed in a study by Fozouni L. et al. [27], similar to our study.

The limitations of the present study were: lack of long-term follow-up for clinical outcomes, single-center cohort, and lack of cohort homogeneity. Despite the limitations, our study adds a notable contribution to the epidemiology of malnutrition/sarcopenia in cirrhotic patients and provides useful information regarding the nutritional assessment among patients with cirrhosis, which is currently lacking, especially in Romania.

In addition to malnutrition/undernutrition, obesity is progressively observed in patients with cirrhosis because of the increasing number of cirrhosis cases related to non-alcoholic steatohepatitis (NASH). Muscle mass depletion may also occur in these patients, but sarcopenia might be overpassed due to the coexistence of obesity. In our study, the prevalence of sarcopenic obesity was high, i.e., 31.7%. In a review conducted by Eslamparast T. et al. [28], the reported prevalence of sarcopenic obesity was between 20% and 35%.

We think that the results of our study are important for daily clinical practice, offering an easy-to-use tool for a better nutritional evaluation of cirrhotic patients, aiming to correct as much as possible malnutrition in this category of patients. Nutritional counseling should be performed in cirrhotic patients with malnutrition/sarcopenia, helping patients to obtain adequate caloric and protein intake.

## 5. Conclusions

This study confirms the high prevalence of malnutrition/sarcopenia in cirrhotic patients. Although CT-based cross-sectional imaging remains the most accurate method for diagnosing sarcopenia/malnutrition, our study highlights the potential use of a simpler and inexpensive alternative that can be used as a valuable tool in daily practice. A model including RFH-NPT and MUAC can be used, with a very good accuracy. Due to the limitations of this study, more research will need to be conducted to support these statements.

## Figures and Tables

**Table 1 diagnostics-12-00893-t001:** Baseline characteristics of the 156 patients studied.

Parameter	Values
Age [years] (mean ± SD) • <40 years • 40–60 years • >60 years	61.8 ± 8.7 0 64 (41%) 92 (59%)
Gender–Men n (%)	96 (61.5%)
Child-Pugh classification	
A B C	34 (21.8%) 61 (39.1%) 61 (39.1%)
Mean Child Pugh score (points)	8.7 ± 2.2
Mean MELD score (points)	14 (19)
Ascites n (%)	
Absent Present	53 (34.0%) 103 (66.0%)
Etiology of cirrhosis n (%)	
Hepatitis B Hepatitis C Alcohol abuse Autoimmune Other	18 (11.5%) 40 (25.6%) 89 (57.1%) 3 (1.9%) 6 (3.9%)
Esophageal varices present—n (%)	104 (66.7%)
Mean BMI (kg/m^2^)	25.9
Underweight—n (%) Normal weight—n (%) Overweight—n (%)	4 (2.7%) 67 (42.9%) 85 (54.4%)
Mean Albumin (g/L ± SD)	2.6 ± 0.7
Mean Hemoglobin level (g/L ± SD)	10.4 ± 2.6

SD, standard deviation; MELD, model for end stage liver disease; BMI, body mass index.

**Table 2 diagnostics-12-00893-t002:** Univariate logistic analysis of factors associated with malnutrition.

Parameter (Reference Category)	Odds Ratio (95% CI)	*p* Value
Age over 60 years	0.92 (0.91–0.99)	0.006
Child-Pugh score *	1.38 (1.18–1.63)	0.0009
MELD score	1.05 (1.00–1.10)	0.01
Child-Pugh class A Child-Pugh class B Child-Pugh class C	0.18 (0.08–0.43) 1.02 (0.53–1.98) 3.50 (1.20–4.25)	0.09 0.05 <0.0001
Lower serum albumin levels *	0.34 (0.20–0.58)	˂0.0001
Vitamin D deficiency *	5.66 (2.18–14.70)	˂0.0001
Gender (male)	3.42 (1.66–7.04)	0.0008
Etiologies		
Alcoholic Hepatitis B Hepatitis C	1.44 (0.75–2.75) 1.06 (0.33–3.40) 0.46 (0.21–0.99)	<0.0001 0.01 0.001

CI: confidence interval; MELD: model for end stage liver disease; * variables were categorized: Child-Pugh score above 7, albumin below 3.4 g/dL, vitamin D below 20 ng/mL.

**Table 3 diagnostics-12-00893-t003:** Sarcopenic diagnostic performance of different nutritional assessment tools considering low skeletal muscle index and low handgrip strength as reference.

Parameter	AUROC	Sensibility (%)	Specificity (%)	Positive Predictive Value (%)	Negative Predictive Value (%)	*p*-Value
RFH-NPT score	0.86 *	76.6	88.7	91.1	71.4	<0.0001
MUAC	0.81	80.8	72.5	81.7	71.4	<0.0001
MUMC	0.79	90.4	58.0	75.2	79.1	<0.0001
SGA score	0.71	81.9	61.2	76.2	69.1	<0.0001
DRY BMI	0.68	42.5	91.9	83.7	50.5	<0.0001
TSF	0.63	41.4	80.6	76.5	47.6	0.002
BMI	0.62	32.9	90.3	83.8	47.1	0.005

RFH-NPT, Royal Free Hospital-Nutritional Prioritizing Tool; MUAC, mid-upper arm circumference; MAMC, mid-arm muscle circumference; SGA, Subjective Global Assessment; DRY BMI, dry body mass index; TSF, skinfold thickness; BMI, body mass index; AUROC, receiver operating curve analysis. * best diagnostic accuracy for the detection of sarcopenia (measured by low SMI + low HGS).

**Table 4 diagnostics-12-00893-t004:** Diagnostic accuracy of several models in the assessment of malnutrition.

Diagnostic Methods	SGA	RFH-NPT	MUAC	MAMC	TSF	HGS
BMI	0.74	0.87	0.81	0.79	0.64	0.16
SGA		0.88	0.83	0.83	0.75	0.41
RFH-NPT			0.89 *	0.89 *	0.87	−0.44
MUAC				0.82	0.82	0.30
MUMC					0.82	0.22
TSF						0.19

Abbreviations: RFH-NPT, Royal Free Hospital-Nutritional Prioritizing Tool; MUAC, mid-upper arm circumference; MAMC, mid-arm muscle circumference; HGS, handgrip strength; SGA, Subjective Global Assessment; TSF, skinfold thickness; BMI, body mass index; * greatest diagnosis accuracy.

**Table 5 diagnostics-12-00893-t005:** The agreement (k-coefficient) between different nutritional assessment tools according to the skeletal muscle index evaluated by Computer Tomography and handgrip strength.

Diagnostic Methods	SGA	RFH-NPT	MUAC	MAMC	TSF	HGS	SMI + HGS
BMI	0.03	0.06	0.06	0.11	0.11	0.05	0.04
SGA		0.45	0.36	0.23	0.30	0.40	0.44
RFH-NPT			0.41	0.32	0.27	−0.44	0.62 *
MUAC				0.54	0.41	0.29	0.47
MUMC					0.12	0.20	0.39
TSF						0.17	0.20

Abbreviations: BMI, body mass index; SGA, Subjective Global Assessment; RFH-NPT, Royal Free Hospital-Nutritional Prioritizing Tool; MUAC, mid-upper arm circumference; MUMC, mid-upper arm muscle circumference; TSF, skinfold thickness; HGS, handgrip strength; SMI (CT), skeletal muscle index evaluated by Computer Tomography. * Strongest agreement.

## Data Availability

Not applicable.

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
