# Peer review of "Comparison of Different Nutritional Assessment Tools in Detecting Malnutrition and Sarcopenia among Cirrhotic Patients"

_diagnostics, 2022, doi:10.3390/diagnostics12040893_

Round 1
Reviewer 1 Report
Dear authors,
I think that the work is of interest to the journal's readers since it focuses on a large sample of specific patients (cirrhotic) and also includes the evaluation of malnutrition by the gold standard methods such as CT.
However, I consider that the statistical analysis contains important conceptual errors and there are results that are not clear how they were obtained, which makes reproducibility difficult. Likewise, there are analyzes that include redundant variables (such as including the MAMC and the two variables used for its calculation, such as the MUAC and the TSF) in the same analysis. All this makes the results unreliable and does not allow a conclusion to be reached on which method would be appropriate to replace the gold-standard in routine follow-up, especially in bedridden patients.
For all that has been described, I believe that the work is not suitable for publication in its current state and that it requires a deep conceptual and statistical review in order to be published.
I enclose the document with my detailed comments that I hope will help you to improve the manuscript.
Best whises,

Author Response
Response to Reviewer #1
Dear Reviewer #1,
Thank you for the review, for your suggestions and for the extensive work. We found your
comments and suggestions extremely usefull. All changes were made in the manuscript with
Track changes function.
Comment #1: When the authors say "malnutrition" they are really referring to undernutrition,
since overweight/obesity is also malnutrition but it is not what is being evaluated here. I think
some mention should be made of the fact that in this type of patient, "malnutrition" is considered
to be malnutrition by default (undernutrition or low weight) and not so much overweight.
Response: We thank the reviewer for his/her comment. Yes, we were referring to undernutrition.
We corrected into the manuscript.
Comment #2: It should be provided the whole term and the acronym in parenthesis after
Response: We thank the reviewer for his/her observation. We corrected into the manuscript.
Comment #3: The whole term should be provided
Response: We thank the reviewer for his/her observation. We corrected into the manuscript.
Comment #4: The paper does not really address these two aspects since no values are reported
for the time spent or cost associated with each type of nutritional assessment. I believe that this
part of the objectives should be eliminated.
Response: We thank the reviewer for his/her observation. We changed into the manuscript.
Comment #5: It would be convenient to provide more location data, at least region/country or,
if possible, even the center's name.
Response: We thank the reviewer for his/her observation. We corrected into the manuscript.
Comment #6: I see that the study period includes the hardest period of the COVID-19 pandemic.
Has this influenced the prevalence figures for cirrhosis seen at the center? As I understand it,
many medical services have been affected by the redirection of human and material resources,
and this has influenced a period of underdiagnosis and underfollow-up of patients affected by
other pathologies. I think it would be appropriate to comment on this aspect in the study's
limitations since, like me, anyone reading these dates may think the same.
Response: We thank the reviewer for his/her observation. Being an emergency hospital and a
tertiary clinic center, our work was not affected by the Covid-19 condition. Our chronic patients
had access to our services, in our ambulatory or even were hospitalized for a period of days,
depending of the patient’s health condition.
Comment #7: It would be convenient to provide bibliographic citations for these diagnostic
methods (for the replicability of the study).
Response: We thank the reviewer for his/her observation. We corrected into the manuscript.
Comment #8: More information would be provided on which specific ethics committee has
assessed the project and the reference number of the approval.
Response: We thank the reviewer for his/her observation. We corrected into the manuscript.
Comment #9: Acronyms can be used since they have been already mentioned before
Response: We thank the reviewer for his/her observation. We corrected into the manuscript.
Comment #10: MUAC
Response: We thank the reviewer for his/her observation. We corrected into the manuscript.
Comment #11: It should be added (BMI) after.
Response: We thank the reviewer for his/her observation. We corrected into the manuscript.
Comment #12: SGA
Response: We thank the reviewer for his/her observation. We corrected into the manuscript.
Comment #13: RFH-NP
Response: We thank the reviewer for his/her observation. We corrected into the manuscript.
Comment #14: BMI
Response: We thank the reviewer for his/her observation. We corrected into the manuscript.
Comment #15: height (m). To avoid confusion with units in cm
Response: We thank the reviewer for his/her observation. We corrected into the manuscript.
Comment #16: Is this a standardized/recommended method by any scientific or clinical society?
The reference supporting this method of procedure should be provided
Response: We thank the reviewer for his/her observation. We corrected into the manuscript.
Comment #17: Acronym only
Response: We thank the reviewer for his/her observation. We corrected into the manuscript.
Comment #18: (TSF)
Response: We thank the reviewer for his/her observation. We corrected into the manuscript.
Comment #19: This can be erased since there is only one equation in the paper
Response: We thank the reviewer for his/her observation. We corrected into the manuscript.
Comment #20: I think there could be a mistake with this cite. In the reference list this is
McDowell et al. and it is a reference for children.
Response: We thank the reviewer for his/her observation. We corrected into the manuscript.
Comment #21: between 5-10th percentiles
Response: We thank the reviewer for his/her observation. We corrected into the manuscript.
Comment #22: < ??
Response: We thank the reviewer for his/her observation. We corrected into the manuscript.
Comment #23: HGS
Response: We thank the reviewer for his/her observation. We corrected into the manuscript.
Comment #24: SMI
Response: We thank the reviewer for his/her observation. We corrected into the manuscript.
Comment #25: area
Response: We thank the reviewer for his/her observation. We corrected into the manuscript.
Comment #26: How were they calculated, by the youden index? Please clarify.
Response: We thank the reviewer for his/her observation. We added the information.
Comment #27: Logistic regression analysis
Response: We thank the reviewer for his/her observation. We corrected into the manuscript.
Comment #28: Dado que más de la muestra presenta sobrepeso, sería relevanete conocer la
prevalencia de obesidad-sarcopénica en esta muestra de pacientes.
Response: We thank the reviewer for his/her observation. Sarcopenic obesity was found in
27/85 (31.7%) of the patients from our cohort, but this was not the aim of this study. We added
the information in the manuscript.
Comment #29: What test has been used here? I understand it is a chi-square but, have two by
two comparisons been made and in all cases has it given p<0.001? The statistical analysis is not
clear
Response: We thank the reviewer for his/her raised question. We used the chi-square test for
independence. We used a 9x2 contingency table- we had 9 groups and two categories and the
result was p<0.0001
Comment #30: What specific malnutrition classification was used for this analysis? Since the
prevalence of malnutrition was analyzed by several methods in the previous section, it is useful
to clarify which of these methods was used to classify patients as malnourished/non-
malnourished in this regression analysis.
Response: We thank the reviewer for his/her raised question. Low muscle strength determined
by handgrip strength and low skeletal muscle index evaluated by Contrast-enhanced CT. We
used their combination as our reference method.
Comment #31: I recommed to write:Parameter (reference category)
Response: We thank the reviewer for his/her observation. We added the information.
Comment #32: Wich is the age range considered as reference category? This is de OR of under
60 years over the category of over 60 years? It should be clearly stated.
Response: We thank the reviewer for his/her raised question. We added the information into
the manuscript. Age over 60 years was used for analysis.
Comment #33: I do not understand how this OR has been calculated. The OR calculation
requires a 2x2 contingency table (two dichotomous variables). It appears that the unclassified
continuous variable (the score) has been used, which would be an error. Please clarify this.
Response: We thank the reviewer for his/her raised question. All continuous variables can be
dichotomised in order to do a statistical analysis. We splatted the variables into lower and
higher. Lower for Class A and higher for class B and C.
Comment #34: Same commentary... How do you categorize albumin and vitamin D to get a
dichotomous variable to perform a logistic regression analysis?
Response: We thank the reviewer for his/her raised question. All continuous variables can be
dichotomised in order to do a statistical analysis. We divided the variables into low variables
(deficiency), and normal values (we didn’t have patients with high levels of vitamin D and
albumin- above the normal values).
Comment #35: typo
Response: We thank the reviewer for his/her observation. We corrected into the manuscript.
Comment #36: The correlation coefficient can only be used to associate two continuous
variables or continuous variables with ordinal variables. Here it seems that they are associating
continuous variables such as the values of anthropometric measurements with a qualitative
categorical variable that would be the presence/absence of malnutrition, which would be
incorrect.
To check for association, it is sufficient to perform a posterior ROC curve analysis or univariate
logistic regression analysis to obtain the regression coefficients.
Response: We thank the reviewer for his/her raised question and for his/her concern. Between
our authors we have a licentiate biostatistician as well, so we decided to report the correlation
coefficient to estimate the strength; We did the Phi correlation and also the point-biserial
correlation. The Point-Biserial Correlation is a special case of the Pearson Correlation and is
used when you want to measure the relationship between a continuous variable and a
dichotomous variable, or one that has two values. Phi represents the correlation between two
dichotomous variables. As with the point-biserial, computing the Pearson correlation for two
dichotomous variables is the same as the phi. The phi is a nonparametric statistic used in cross-
tabulated table data where both variables are dichotomous.
Comment #37: Not clear what this p-value measures...
Response: We thank the reviewer for his/her raised question. This p value is the result between
the analysis of all Roc curves.
Comment #38: Sarcopenic diagnostic performance??
Response: We thank the reviewer for his/her observation. We corrected into the manuscript.
Comment #39: RFH-NPT score
Response: We thank the reviewer for his/her observation. We corrected into the manuscript.
Comment #40: This variable should not be included in the analysis since it is necessary to
calculate the dependent variable (handgrip strength).
Response: We thank the reviewer for his/her observation. We corrected into the manuscript.
Comment #41: SGA score
Response: We thank the reviewer for his/her observation. We corrected into the manuscript.
Comment #42: This would actually be: *best diagnostic accuracy for the detection of sarcopenia
(measured by low SMI+low HGS)
Response: We thank the reviewer for his/her observation. We corrected into the manuscript.
Comment #43: It is not at all clear how this analysis was performed. I understand that what is
shown in the table are AUCROC values and that the dependent variable is still presence of
sarcopenia measured as low SMI+ low HGS? If so, how have two continuous variables been
combined for this diagnosis?
Please clarify in more detail how this analysis has been performed (here or in the methodology
section).
Response: We thank the reviewer for his/her raised question. Yes, in table 4 were presented the
AUCs. The two variables were combined same as in the previous analysis, table 3. When
analysed, a new variable was made with the name
“sarcopenia” and this new variable was a dichotomous one. If we describe all the statistical
steps for this kind of analysis, we will load the manuscript with technical/statistical data that is
unnecessary for a regular reader.
Comment #44: Since the MUAC value is needed to calculate the MAMC, this analysis is
redundant.... Since the MUAC has better diagnostic accuracy than the MAMC, it would not
make sense to use it since it is more complex to obtain (it requires including the TSF and making
a calculation, while the MUAC is ua has a single cut-off point). For this same reason, in Table
4 it does not make sense to make combinations of MUAC-MAMC or TSF-MAMC since they
are variables included in their own calculation. In any case, it would make sense to compare
the diagnostic efficacy of MUAM-TSF with that of MAMC.
Response: We thank the reviewer for his/her raised question. MUAC did not have a better
diagnostic accuracy than MAMC. It is the same diagnostic accuracy as showed in table 4, that
is why we decided to show all this information, even the fact that MUAC is needed in order to
calculate the MAMC.
Comment #45: The Kappa coefficient is used to see the degree of agreement between
classifications... (i.e., categorical variables of presence/absence of disease). Here it seems that
the variables are being analyzed as continuous measures (value of the anthropometric variable
or the score obtained in each patient). Each variable should be classified (e.g. BMI classified as
malnutrition yes/no) and compared with the sarcopenia classification obtained as
lowSMI+lowHGS).
Response: We thank the reviewer for his/her raised question. We did not take the variables as
a continuous one. Our variables were classified as a categorical variable in order to do this
analysis, to calculate the Kappa coefficient.
Comment #46: As I commented earlier, given that more than half of the sample is overweight,
it is worth assessing the prevalence of sarcopenic obesity and reviewing the literature on its
possible effect on patients with liver cirrhosis.
Response: We thank the reviewer for his/her suggestion. We mentioned the prevalence in the
anterior comment #28.
Comment #47: Aa expected why? Authors should explain the reason (failure to assess body
composition)
Response: We thank the reviewer for his/her raised question. In the literature, the performance
of BMI in detecting malnourished patients is low, and it was not a surprise to find that our
results are similar to the literature, but we agree that the use of that expression can lead to
interpretations, so we corrected into the manuscript.
Comment #48: Erase name, it should be mentioned only the last name et al.
Response: We thank the reviewer for his/her observation. We corrected into the manuscript.
Comment #49: erase
Response: We thank the reviewer for his/her observation. We corrected into the manuscript.
Comment #50: What type of population studied those authors? It is important to know if the
population are comparable
Response: We thank the reviewer for his/her raised question. Both mentioned studies used
caucasian patients with cirrhosis in different stages, with similar mean age.
Comment #51: erase
Response: We thank the reviewer for his/her observation. We corrected into the manuscript.
Comment #52: Acronym
Response: We thank the reviewer for his/her observation. We corrected into the manuscript.
Comment #53: It would be useful to add here or in the introduction, what kind of actions are
put in place when malnutrition/sarcopenia is diagnosed in patients with cirrhosis to avoid the
risk of death.
Response: We thank the reviewer for his/her suggestion We added this to the manuscript.
Comment #54: The authors do not make clear, after all the analyses, which method(s) they
would choose on a day-to-day basis to assess the risk of malnutrition/sarcopenia in their
patients.
Response: We thank the reviewer for his/her suggestion. We added the information in order to
be more understandable.
Reviewer 2 Report
- Important considerations for the authors:
- Standardize the name. Is “Romania” or “Romania” the correct form ?
- The template needs to be revised. The authors did not follow the template. There are points to be reviewed. Words with another color, underlined. This needs to be corrected as it seems that the article is still in the elaboration phase.
- The methodology is very simple. There is no more refined molecular marker in the methodology.
- Some phrases are without reference. See lines 49; 69
- In the material and methods section, insert the ethics committee approval number;
- In equations, use appropriate tools to create equations;
- Some measurement units are in bold. Please make the correction;
- The format of the tables is incorrect. Check the template.
- English needs to be improved.
Author Response
Response to Reviewer #2
Dear Reviewer #2,
Thank you again for the review and suggestions. We found your comments and suggestions
extremely useful. All changes were made in the manuscript with Track changes function.
Important considerations for the authors:
Comment #1: Standardize the name. Is “Romania” or “Romania” the correct form ?
Response: We thank the reviewer for his/her observation. We corrected into the manuscript. It
is “Romania”
Comment #2: The template needs to be revised. The authors did not follow the template. There
are points to be reviewed. Words with another color, underlined. This needs to be corrected as
it seems that the article is still in the elaboration phase.
Response: We thank the reviewer for his/her observation. We corrected.
Comment #3: The methodology is very simple. There is no more refined molecular marker in
the methodology.
Response:We thank the reviewer for his/her observation. Yes, we did not include any molecular
marker, because the aim of our study was to compare the performance of different nutritional
assessment techniques in detecting malnourished cirrhotic patients.
Comment #4: Some phrases are without reference. See lines 49; 69
Response: We thank the reviewer for his/her observation. We corrected into the manuscript.
Comment #5: In the material and methods section, insert the ethics committee approval number;
Response: We thank the reviewer for his/her observation. We corrected into the manuscript.
Comment #6: In equations, use appropriate tools to create equations;
Response: We thank the reviewer for his/her observation. We corrected into the manuscript.
Comment #7: Some measurement units are in bold. Please make the correction;
Response: We thank the reviewer for his/her observation. We corrected into the manuscript.
Comment #8: The format of the tables is incorrect. Check the template.
Response: We thank the reviewer for his/her observation. We corrected into the manuscript.
Comment #9: English needs to be improved.
Response: We thank the reviewer for his/her observation and suggestion. We gave to a native
English writer to correct the English.
Round 2
Reviewer 1 Report
Dear Authors,
Thank you for taking my comments into account. I think the work has improved from the previous version. However, I believe that there is still room for improvement in aspects related to the treatment of data for statistical analysis. The methodology section should be improved/expanded by describing in more detail some of the tests used and it should be indicated in the regression and concordance analyses how those variables that are continuous have been converted into dichotomous variables. It is not clear reading the work in its current state and I believe that these are reasonable doubts that will arise to any reader with knowledge of statistics. Reproducibility must also be assured.
I leave my more detailed comments below.
Kind regards
- In the formula, it is written MUMC but in the text MAMC. Please unify this.
- Comment #33 and #34: Authors said "All continuous variables can be
dichotomised in order to do a statistical analysis. We divided the variables into low variables (deficiency), and normal values (we didn’t have patients with high levels of vitamin D and albumin- above the normal values).". It is good that they explain it to me in the answers but this should also be clear for future readers of the work, so the categories created within the table should be indicated indicating the reference category or indicate in the methodology how these variables have been classified (what cut-off points have been considered to consider a good and bad level). - Comment 36: Authors said "Between our authors we have a licentiate biostatistician as well, so we decided to report the correlation coefficient to estimate the strength; We did the Phi correlation and also the point-biserial correlation. The Point-Biserial Correlation is a special case of the Pearson Correlation and is used when you want to measure the relationship between a continuous variable and a dichotomous variable, or one that has two values. Phi represents the correlation between two dichotomous variables. As with the point-biserial, computing the Pearson correlation for two dichotomous variables is the same as the phi. The phi is a nonparametric statistic used in cross-tabulated table data where both variables are dichotomous." Ok, but this should be clearly stated in the methodology section (Statistical analysis section) to avoid readers' confusion (as it has happened to me).
- Comment 43: I dissent with the answer. I think at least a brief mention should be made in the methodology or results as to how those two variables have been "merged" to be a single continuous variable (what calculation has been done with their individual scores?). This is essential to ensure the replicability of the study, which is one of the main indicators of scientific quality.
- #Commet 45: Authors said "We did not take the variables as a continuous one. Our variables were classified as a categorical variable in order to do this analysis, to calculate the Kappa coefficient." Ok, but it should be indicated somewhere (methodology or the table itself) how each variable has been categorized. BMI < 18.5 kg/m2? the SGA below/above what specific value? what cut-off point for TSF? were standards used or categorized from the median of the sample itself? All this information should be clear for the reproducibility of the study.
Author Response
Dear Reviewer #1,
Thank you for your extensive work and review. We found your comments extremely helpful.
All of the following corrections were revised as requested and were marked with Track Changes in the manuscript. The manuscript was improved according to your suggestions. Please see the attachment

Reviewer 2 Report
The authors made the requested adjustments.
The article may be accepted for publication.
Author Response
Dear Reviewer #2,
Thank you again for the review and suggestions.
Round 3
Reviewer 1 Report
Dear Authors,
I thank you for the effort made to adapt the manuscript to my suggestions and comments. I believe that the work has gained strength and replicability and its results can be used by health personnel in other institutions to improve the diagnosis of malnutrition in their patients.
In my opinion, the work is now ready for publication. Congratulations!
Best wishes,